# Changes in emotional distress among Ontario education workers during the COVID-19 pandemic: 2021–2023

Iris Gutmanis[1], Brenda L. Coleman[1,2]*, Kelly Ramsay[3], Robert Maunder[1,2], Susan J. Bondy[2], Kailey Fischer[1], Veronica Zhu[1], Allison McGeer[1,2]

**1** Sinai Health System, Toronto, ON, Canada, **2** University of Toronto, Toronto, ON, Canada, **3** York University, Toronto, ON, Canada

These authors contributed equally to this work as co-first authors

* b.coleman@utoronto.ca

## Abstract

### Introduction

Education workers experienced increased stress during the COVID-19 pandemic adapting to changing work locations, workload, and pedagogical approaches as well as dealing with pandemic-induced personal life stress.

### Methods

The goal of this Canadian prospective cohort study was to determine whether levels of distress, as measured by the Kessler Psychological Distress Scale (K10), varied significantly over the course of the study (February 18, 2021 to December 22, 2023) among Ontario education workers after adjusting for demographic, work-related, and temporal factors. Exposure data were collected at enrollment and updated annually while vaccination and illness surveys were completed as needed. The K10 was completed periodically throughout the study. Linear mixed effects models were used to assess factors associated with changes in K10 scores.

### Results

On average, K10 scores fell 0.5% every four weeks over the 34-month long study. However, the mean score, that decreased from 23.1 at study start to 19.3 by study end, remained above the cut-off of 15, indicating no ongoing emotional distress. Lower distress scores were also associated with non-winter seasons, older age, being male, and being in very good/excellent health. Alternatively, higher K10 scores were associated with being on medications to reduce anxiety, depression, or sleeping problems, return-to-workplace periods after school lockdowns, and periods of intense non-pharmacological interventions.

**Data availability statement:** The datasets generated and/or analysed during the current study are not publicly available as the data file contains potentially identifying patient information. Requests for data access should be sent to: Dr. Brenda L. Coleman, PhD Clinical Scientist, Sinai Health, Toronto; Assistant Professor, Dalla Lana School of Public Health, University of Toronto b.coleman@utoronto.ca Alternate contact: director@lunenfeld.ca.

**Funding:** This work was funded by: Public Health Agency of Canada (grant number 2021-HQ-000149) (received award: BLC) https://www.canada.ca/en/public-health.html Canadian Institutes of Health Research (grant number 181116) (received award: BLC) https://webapps.cihr-irsc.gc.ca/decisions/p/project_details.html?applId=464063&lang=en. None of the authors received salary or other funding from commercial companies. The funders had no role in study design, data collection and analysis, decision to publish, or preparation of the manuscript.

**Competing interests:** The authors have declared that no competing interests exist.

## Conclusion

Early identification of people most likely to experience distress is needed so that stress remediation strategies can be quickly implemented. Younger, female education workers with lower rated subjective health, who are taking medications to reduce anxiety, depression, or sleeping problems are likely to be at the highest risk. Recognizing periods that intensify stressful situations, such as pandemics and intervals within them, is important to anticipate the need for assistance.

## Introduction

Even prior to the COVID-19 pandemic, work in the education field was identified as stressful [1] with high levels of stress and burnout (e.g., [2,3]). During the COVID-19 pandemic, educators were subject to stressful work-life changes including school lockdowns with associated rapid changes to pedagogical approaches to learning and implementation/enforcement of safety measures such as indoor masking when students returned to in-person learning, and exposure to unvaccinated and ill school staff and students. As youth may have fewer stress reduction resources [4], much of the current research focuses on the impact of the pandemic on students (e.g., [5–7]). However, education workers' well-being is closely linked to instructional quality and students' educational outcomes [8]. Further, prolonged and/or severe stress can lead to distress and increased vulnerability to anxiety and depression [9], resulting in increased time-off work and decreased productivity [10] that can impact student outcomes.

Earlier studies of education workers, who are often recognized as essential workers (e.g., [11,12]), focused on the identification of risk factors associated with emotional distress with the goal of early identification of those most likely to benefit from stress-relief strategies. Some studies specific to education workers found that emotional distress was greater for females [13–17]; however, results have been inconsistent with some studies finding no differences by sex [18,19]. Similarly, while most studies found that COVID-19 associated emotional distress decreased with age [14,17], such findings were not always replicated [19,20]. Additional personal- and work-related factors associated with higher levels of distress include living alone [17,21], COVID-19 [22], receiving a vaccine (vs. not vaccinated) [23], working in an elementary (vs. secondary) school [15], more contact time with students [14], part-time (vs. full time) work [17], and transitioning to working from home [24]. Factors associated with decreased distress include better subjective health [16], living in a rural (vs. urban) setting [15], and longer teaching experience [25].

Building on evidence generated from cross-sectional risk-factor studies, longitudinal qualitative [26] and quantitative [27] studies found education worker mental health deteriorated over the course of the pandemic. To provide more detail as to when and why mental health changed over the course of the pandemic, Steigleder at al. surveyed German pre-school teachers monthly [28]. These researchers found that between September 28, 2020 and February 28, 2021, during a period of nationwide

lockdowns, teachers experienced a significant decrease in positive affect as well as an increase in negative affect. Then, during the summer of 2021, positive affect significantly increased while negative affect decreased but the trend reversed when schools reopened in the fall.

Fluctuations in downward trends in emotional distress levels over the course of the COVID-19 pandemic have also been observed in other essential or frontline workers. For example, between 2021 and 2023, Kessler Psychological Distress Scale scores [29] decreased over time among Canadian healthcare providers. However, the rate of decline in emotional distress scores decreased with increased intensity of public health restrictions, during the winter season, and if the healthcare provider was taking antidepression, anti-anxiety or anti-insomnia medications [30].

The goal of this exploratory study was to determine if distress, as measured by the 10-item Kessler Psychological Distress Scale (K10), varied significantly over the course of the study (February 18, 2021 to December 22, 2023) among Ontario education workers after adjusting for demographic, work, and temporal factors. We hypothesized that K10 scores would significantly decrease over time during the COVID-19 pandemic after adjusting for factors known to impact distress. Further, we hypothesized that the rate of decline would decrease during periods of intense public health restrictions and during the first four weeks after returning to the workplace following periods of school lockdowns but would increase during summer vacation. We also assessed the change in the K10 subscale scores for anxiety and depression.

## Materials and methods

This observational cohort sub-study was embedded within the 34-month COVID-19 Cohort Study of Teachers and Education Workers, a study designed to identify the incidence and risk factors for infection with COVID-19 (for study details see Coleman et al. [31]). On February 18, 2021, once funding was secured, the study began enrolling education workers, aged 18–77 years, employed in an Ontario school or school board, working an average of ≥8 hrs/week and planning to continue working for at least three months. Initially planned as a one-year study, additional funding was secured to allow for data collection to the end of 2023. Rolling enrollment continued until June 1, 2023 with data collection ending December 22, 2023. Additional study funding was not sought since the World Health Organization (WHO) declared the end of the public health emergency of international concern on May 5, 2023 [32]. Education workers were eligible for this sub-study if they submitted one or more K10 surveys. The ten individuals who indicated their gender was neither male nor female were dropped from the analysis as there were too few individuals to draw any meaningful conclusions.

### Survey instruments

Between February 2021 and August 2022, participants were asked to complete a K10 survey upon enrollment and then every three months. Subsequently, to reduce respondent burden participants completed the K10 every six months. The K10, a widely used screening tool of non-specific psychological distress [33], with well established psychometric properties, provides scores ranging from 10 to 50, with higher scores indicating greater distress. These analyses use a score of ≥16 to identify those most likely to be experiencing distress [34] with scores also categorized as low (<16 points), moderate (16–21), high (22–29), and very high (30–50) [35]. Depression (range: 6–30) and anxiety (range: 4–20) sub-scale scores were also calculated as per Brooks et al. [36].

Baseline surveys were completed at enrollment and updated annually. Participants were also asked to submit information whenever they received a COVID-19 vaccination and complete an illness survey whenever they were tested for COVID-19 or experienced symptoms of an acute respiratory illness. Receipt of COVID-19 vaccine or a positive test result (polymerase chain reaction or rapid antigen) within six months prior to each K10 were used in these analyses. Age, gender, postal district, subjective health ("In general, would you say your health is…"), and use of anti-anxiety, antidepressants, and anti-insomnia medications ("Do you take any other prescription medications, not including birth control? Specify.") were taken from the baseline survey completed at enrollment. Other demographic variables (e.g., people per bedroom; a composite variable created from number of people in household and number of bedrooms in house) and

occupational variables (e.g., occupation, level of student or coworker contact [4 categories: no close contact, same room/space but rarely within 2 metres, same room/space and often within 2 metres, same room/space and physical contact], number of weekly student contacts ["In an average week, with how many different students do you have close, extended contact, within 2 metres and for 2 minutes or longer?"]) were taken from the baseline survey completed closest to, but preceding, each K10 submission.

Four temporal factors were assessed for their possible impact over the course of the study. Time was measured in four-week periods to correspond with the recall period stipulated in the K10 questionnaire ("In the past four weeks, about how often did you feel …"), while the four seasons correspond to the solstices and equinoxes. In Ontario, both elementary and secondary schools were closed to in-person instruction four times during the pandemic; twice during this study period: April 15, 2021 to September 6, 2021 and December 17, 2021 to January 16, 2022. As education workers' concerns regarding return to their workplace following periods of school lockdowns have been noted [37], two 4-week return-to-workplace periods were identified to assess possible associated changes in distress levels: September 7, 2021 to October 4, 2021 and January 17, 2022 to February 13, 2022. Mitigation periods were based on COVID-19 Policy Response Canadian tracker data [38]. The intensity of non-pharmacological intervention strategies pertaining to three sectors (work, school, and other locations) were graded on a four-point ordinal scale (0–3 [most restrictive]) (e.g., level 2: school sector: some schools and childcare centres closed, or combination of online and in-person classes implemented; work sector: working from home strongly suggested or most businesses closed except for specific sectors or categories of workers; or other sectors: stricter public gathering restrictions, some travel restrictions between provinces, closure or significantly reduced capacity of most indoor activities, closure of some outdoor activities). Periods in which the three sectors' scores summed to ≥7 were considered high mitigation.

## Data analysis

Frequency distributions were generated for categorical variables and measures of central tendency for all continuous variables. Univariable analyses were done in Stata SE (v.18.0). The continuous variables were normalized to the interval [0,1] before estimating the model parameters. To adjust for heteroscedasticity, K10 scores were logarithmically transformed in all linear mixed effects models. As such, all model estimates are interpreted as the percent change in K10 scores every four-weeks per one point change in the independent variable.

To account for the dependence between observations from the same participants, linear mixed effects models, specifically random intercept models, were fitted to determine the impact of the four study temporal variables on K10 scores. A linear mixed effects explanatory model that included all significant temporal factors, demographic and health factors (age, gender, people per bedroom, use of anti-anxiety, antidepression or anti-insomnia prescription medications, subjective health, postal district), occupational factors (occupation, level of student and co-worker contact), and COVID-19-specific factors (COVID-19 within past six months, COVID-19 vaccination within past six months) was fitted. The most parsimonious model was chosen via backward elimination using the p-value for the F-test as the determinant of retention, with a threshold of 0.05. If all levels of a categorical variable together explained a significant amount of variation in the K10 scores, that variable was retained. Overall model fit, heterogeneity of variance assumptions, normality assumptions, and time dependence were all assessed to ensure model assumptions were not violated. Models were fit using the lme4 package in R software [39].

A sensitivity analysis was conducted to assess whether estimates were similar for participants who were with the study for shorter versus longer periods; we used the same model-building parameters described above but excluded data from respondents who submitted fewer than five K10 surveys. Relationships between the depression and anxiety sub-scale scores and study covariates were also assessed using the methods described above.

The study was approved by Sinai Health Research Ethics Board (IRB: 20–0343-A) and conducted in accordance with standards established in the declaration of Helsinki. Informed consent was obtained from all participants involved in the study prior to any data collection.

## Results

Three thousand five hundred three people submitted 14,224 K10 surveys between February18, 2021 and December 22, 2023. As seen in Table 1, more than 80% of participants were teachers, the median age was 45.4 years (interquartile range (IQR) 39.4, 51.7) and the median household size was 4 (IQR 2, 4). Almost one quarter of participants had direct contact with students while only 14.6% were male, 15.6% were taking anti-anxiety, antidepressant, or anti-insomnia medications at enrollment, and 3.1% had experienced COVID-19 in the six months before completing their first K10.

The mean K10 score during the first four weeks of the study (submitted Feburary18, 2021 to March 21, 2021) was 23.1 (95% confidence interval (CI) 22.7, 23.6) while the final eight weeks (October 30, 2023 to December 22, 2023) had a mean score of 19.3 (CI 18.8, 19.8). At study start, 141 of 940 (15.0%) participants were categorized as having a low level of distress, 30.3% moderate, 35.4% high, and 19.3% very high. By November 2023, the percent of participants with a low level of distress increased to 34.0% (236/694) while 33.4% had moderate, 24.3% had high, and 8.2% had very high K10 scores.

In adjusted linear mixed effect regression models, all four temporal factors were significantly associated with distress. Specifically, there was an average 0.5% decrease in K10 scores every four-weeks throughout the study (Fig 1 and Table 2). Compared to winter periods, spring periods were associated with a 1.3% decrease in K10 scores. Summers were associated with a 12.9% decrease in K10 scores that were followed by a rebound in scores in the autumn. The return-to-workplace and high mitigation periods were associated with a 4.4% and 2.1% increase in scores, respectively.

Lower K10 scores were significantly associated with older age, being male, being in very good/excellent subjective health, and living in postal districts with lower population densities. Higher scores were associated with being on medications to reduce anxiety, depression, or sleeping problems and with having received a COVID-19 vaccine in the six months prior to completing the K10. Although, no single element was significantly different from the referent category, the "highest level of student contact" variable was also associated with K10 scores. The model-conditional coefficient of determination ($R^2$), estimating the goodness of fit for both the fixed and random effects, was 65.7% and the marginal $R^2$, measuring only the fixed effects (those that are the same for all participants), was 14.9%.

### Sensitivity analysis

Among those submitting five or more K10s (8693 observations among 1369 participants), mean scores were similar to those for all participants; they changed from 22.9 (CI 22.3, 23.6) in February/March 2021 (low risk: 74/446 or 16.6%) to 19.2 (CI 18.6, 19.8) in November/December 2023 (low risk: 205/556 or 36.9%). The linear mixed effects model was also largely similar to the one estimated using all responses (see S1 Table).

### Anxiety and depression scores of K10

Similar to the K10 model, both anxiety and depression scores decreased over the course of the study. Although all non-winter seasons were associated with lower depression scores compared to winter periods, only the summer periods were associated with significantly lower anxiety scores. The return-to-workplace periods were associated with higher scores on both the anxiety and depression sub-scales, while periods of high mitigation were associated with higher anxiety scores only.

As seen in Table 3, decreases in both anxiety and depression scores were associated with older age, being male, and being in very good/excellent subjective health after adjusting for temporal factors. Higher scores were associated with being on anti-anxiety, antidepression or anti-insomnia medications and receiving a COVID-19 vaccine sometime in the six months prior to K10 completion. Factors associated with higher anxiety sub-scale scores only, included having had COVID-19 in the previous six months and having physical contact with coworkers. Meanwhile, administrative and clerical staff reported significantly lower anxiety scores than teachers while educational assistants, early childhood educators,

**Table 1. Characteristics of Ontario education workers of the COVID-19 Cohort Study who participated in the K10 sub-study (February 18, 2021 to December 22, 2023): Number (%) unless otherwise stated.**

| Characteristic | Participants (N = 3503) |
|---|---|
| Age, in years, median (IQR) | 45.4 (39.4, 51.7) |
| Gender: | |
| Female | 2989 (85.3) |
| Male | 514 (14.7) |
| Number of people per bedroom, median (IQR) | 1 (0.8, 1.3) |
| Anti-anxiety, antidepression or anti-insomnia medications | |
| No | 2956 (84.4) |
| Yes | 547 (15.6) |
| Subjective health | |
| Poor/fair/good | 1065 (30.4) |
| Very good/excellent | 2438 (69.6) |
| Postal district | |
| Central Ontario | 1209 (34.5) |
| Eastern Ontario | 624 (17.8) |
| Metropolitan Toronto | 698 (19.9) |
| Southwestern Ontario | 797 (22.8) |
| Northern Ontario | 175 (5.0) |
| COVID-19 vaccine within six months of the first K10 | |
| Yes | 2072 (59.1) |
| No | 1431 (40.9) |
| COVID-19 prior to the first K10 | |
| Yes | 198 (5.6) |
| No | 3305 (94.4) |
| Occupation | |
| Teacher/instructor | 2832 (80.8) |
| Educational assistant/ECE | 281 (8.0) |
| Principal/vice principal | 131 (3.7) |
| Administrative/clerical[1] | 91 (2.6) |
| Other, professional support[2] | 123 (3.5) |
| Other, operational support[3] | 45 (1.3) |
| Level of student contact | |
| None/not same room | 243 (6.9) |
| Same room, rarely <2 metres (~6 feet) | 560 (16.0) |
| Same room, often <2 metres | 1876 (53.6) |
| Same room, physical contact | 824 (23.5) |
| Level of coworker contact | |
| None/not same room | 316 (9.0) |
| Same room, rarely <2 metres | 1427 (40.7) |
| Same room, often <2 metres | 1595 (45.5) |
| Same room, physical contact | 165 (4.7) |
| Number student contacts/week (including 0), median (IQR) | 23 (10, 46) |

IQR: interquartile range; ECE: Early childhood educator; K10: 10-item Kessler Psychological Distress Scale.

[1] office and clerical staff, superintendent, human resources, finance, planner.

[2] psychologist, social worker, therapist, librarian, registered nurse.

[3] information technology, audio-visual or computer technicians, bus driver, custodian, building maintenance (carpenter, plumber), cafeteria staff, lunchroom assistant.

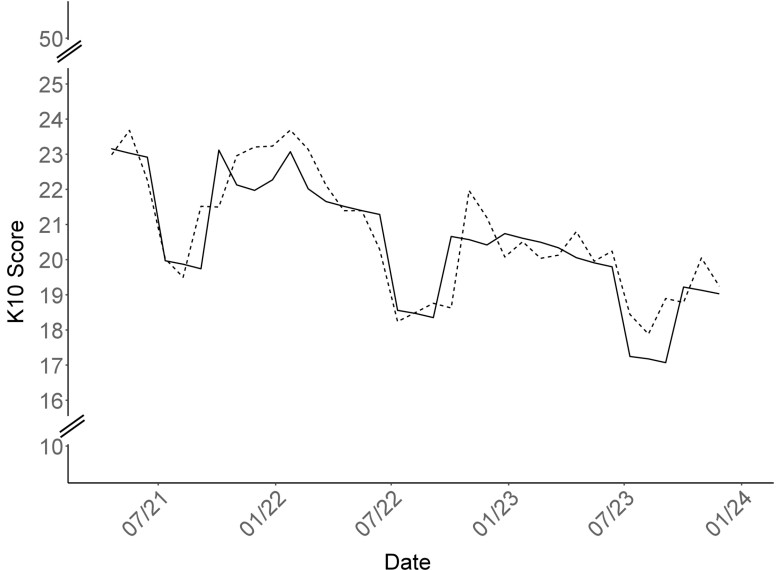

**Fig 1. 10-item Kessler Psychological Distress Scale scores over time, Ontario education workers, COVID-19 Cohort Study, February 18, 2021-December 22, 2023. Mean score: raw mean K10 scores (dotted line).** Fitted score: Predicted K10 scores as estimated by linear mixed effects model adjusted for time, season, and return-to-workplace (solid line).

and operational staff reported significantly lower depression scores than teachers after adjusting for other variables in the model.

## Discussion

In our population of Canadian education workers who completed K10 questionnaires recurrently between February 2021 and December 2023, average distress scores fell by 3.8 points (23.1 to 19.3) but were still above the selected cut-off indicating some emotional distress (≥16). Although distress scores generally declined during the study, scores were higher during periods of high non-pharmacological interventions, and when education workers initially returned to their workplace after school lockdowns; meanwhile scores decreased in non-winter seasons. Older age, male gender, being in very good/ excellent self-reported health, and living in the less densely populated regions of the province (as measured by the postal districts) were associated with lower K10 scores. Meanwhile, using anti-anxiety, antidepression, and/or anti-insomnia medications and recent receipt of COVID-19 vaccines were associated with higher distress scores. Variables associated with overall K10 scores were substantially similar to those associated with the anxiety and depression sub-scale scores, with some differences in effect sizes.

During the COVID-19 pandemic work-related stress and anxiety were noted across the globe; however, impacts were sector and time specific. For example, a cross-sectional study conducted in the spring of 2020, prior to the start of this study, determined that fewer Canadian healthcare providers self-reported symptoms of moderate or high stress, anxiety, and depression than other working adults [40]. Similarly, Canadians who considered themselves essential workers (i.e., healthcare, food services, etc.) reported fewer symptoms of anxiety and depression than non-essential workers when surveyed between May and July 2020 [41]. Also, Bu et al. [42] found that non-essential workers, health/social care providers, and teachers/childcare workers had consistently lower levels of depressive and anxiety symptoms than essential service workers (e.g., public safety workers) in England between March 2020 and February 2021. In contrast, an American study conducted between September 2020 and March 2021 found that teachers reported significantly higher levels of anxiety and depression symptoms than healthcare providers [43]. Likewise, a longitudinal Canadian study found that the percent

**Table 2. 10-item Kessler Psychological Distress Scale scores for Ontario education workers participating in the COVID-19 Cohort Study, February 18, 2021 to December 22, 2023, estimates using a linear mixed effects model.**

| Variable | Percent change per 4-week period[1] (95% CI) |
|---|---|
| Per 4 weeks | **−0.5 (−0.5, −0.4)** |
| Season: | |
| Winter | Referent |
| Autumn | **−1.5 (−2.7, −0.4)** |
| Spring | **−1.3 (−2.4, −0.2)** |
| Summer | **−12.9 (−13.8, −12.0)** |
| Return-to workplace period: | |
| No | Referent |
| Yes | **4.4 (2.8, 6.1)** |
| Mitigation level: | |
| Low | Referent |
| High | **2.1 (0.7, 3.6)** |
| Age, in years | **−0.6 (−0.7, −0.5)** |
| Gender: | |
| Female | Referent |
| Male | **−8.2 (−10.6, −5.7)** |
| People per bedroom | NA |
| Anti-anxiety, antidepression or anti-insomnia medication: | |
| No | Referent |
| Yes | **11.0 (8.1, 14.0)** |
| Subjective health: | |
| Poor/fair/good | Referent |
| Very good/excellent | **−13.3 (−15.0, −11.4)** |
| Postal District: | |
| Central | Referent |
| Eastern | **−3.5 (−6.1, −0.8)** |
| Metropolitan Toronto | −1.8 (−4.3, 0.8) |
| Northern | **−4.7 (−8.9, −0.3)** |
| Southwestern | **−4.5 (−6.9, −2.0)** |
| COVID-19 vaccine in last six months: | |
| None | Referent |
| One or more | **1.4 (0.4, 2.3)** |
| COVID-19 in last six months: | |
| None | Referent |
| One or more | NA |
| Occupation: | |
| Teacher/instructor | Referent |
| Administrative/clerical[2] | NA |
| Educational assistant/ECE | |
| Other, operational support[3] | |
| Principal/VP | |
| Other, professional support[4] | |
| Highest level of student contact: | |

*(Continued)*

**Table 2.** (Continued)

| Variable | Percent change per 4-week period[1] (95% CI) |
|---|---|
| None | Referent |
| In room, often <2 metres | 0.7 (−1.9, 3.3) |
| In room, rarely <2 metres | −0.8 (−3.6, 2.1) |
| Physical contact | 2.1 (−0.6, 5.0) |
| Highest level of coworker contact: | |
| None | Referent |
| In room, often <2 metres | NA |
| In room, rarely <2 metres | |
| Physical contact | |
| Number of student contacts per week | NA |

Bold: significantly different (p < 0.05) from referent group.

ECE: early childhood educator; NA: not applicable (not included in model).

[1] Conditional $R^2$: 65.7%; Marginal $R^2$: 14.9%.

[2] Office and clerical staff, superintendent, human resources, finance, planner.

[3] Information technology, audio-visual or computer technicians, bus driver, custodian, building maintenance (carpenter, plumber), cafeteria staff, lunchroom assistant.

[4] Psychologist, social worker, therapist, librarian, registered nurse.

of healthcare providers with K10 scores indicative of high/very high distress (K10 > 21) was 37.8% in 2021 and 23.1% in 2023 [30], lower than what was found in the current study (2021: 54.7%; 2023: 32.5%) but, in both cases, much higher than what is expected in the general population [44].

The finding, that education workers may have, at times, experienced higher levels of distress than healthcare providers warrants further consideration. A qualitative study conducted in the fall of 2020 among Ontario education workers revealed that teachers were challenged to quickly change their work environment to provide their services using new technologies with limited training opportunities and resources, resulting in increased reports of stress [45]. When in-person education resumed, teachers were concerned about having to work in an environment with less than ideal measures to reduce the spread of disease [46]. Furthermore, the intense public and political scrutiny surrounding school closures and re-openings placed educators at the center of a contentious societal debate [47]. These likely created a unique and potent combination of stressors not experienced in the same way by other worker groups.

Other studies have noted pandemic-related changes in educators' mental health status, but none have followed education workers for as long as the current study (almost three years) or included temporal measures. Zaleski found that well-being scores of American educators decreased between 2019 and 2020 and again between 2020 and 2021 [27]. Among Danish teachers, perceived stress levels dropped between May and June 2020 but then increased in November/December 2020 to above May 2020 levels [48]. As noted above, we found that although distress scores generally declined as the pandemic continued, there were periods of flux including season, mitigation intensity, and return-to-workplace periods.

In the current study, distress levels were lower during Ontario education worker vacation periods (summers). The marked fluctuations observed in Fig 1, particularly the sharp decrease in distress during the summer and the subsequent spike upon returning to the workplace, highlight the significant and immediate impact of work-related environmental factors on emotional well-being. This corroborates previous research that reported a strong, but limited, positive effect of vacations on mood. Among British teachers who took short midterm vacations, anxiety, depression, and emotional exhaustion decreased during the vacation, but levels of emotional distress returned to prevacation levels within four

**Table 3.** Anxiety and depression sub-scale scores from the 10-item Kessler Psychological Distress Scale for Ontario education workers participating in the COVID-19 Cohort Study, February 18, 2021 to December 22, 2023, estimates using linear mixed effects models.

| Variable | Anxiety score estimates[1] (95% CI) | Depression score estimates[2] (95% CI) |
|---|---|---|
| Per 4 weeks | **−0.4 (−0.5, −0.4)** | **−0.5 (−0.5, −0.4)** |
| Season: | | |
| Winter | Referent | Referent |
| Autumn | −0.4 (−1.7, 0.9) | **−2.4 (−3.6, −1.2)** |
| Spring | −1.0 (−2.2, 0.2) | **−1.6 (−2.7, −0.4)** |
| Summer | **−8.4 (−9.5, −7.3)** | **−15.9 (−16.9, −15.0)** |
| Return-to-workplace period: | | |
| No | Referent | Referent |
| Yes | **6.4 (4.5, 8.3)** | **2.9 (1.2, 4.6)** |
| Mitigation level: | | |
| Low | Referent | Referent |
| High | **4.7 (3.0, 6.3)** | NA |
| Age, in years | **−0.7 (−0.8, −0.6)** | **−0.5 (−0.6, −0.4)** |
| Gender: | | |
| Female | Referent | Referent |
| Male | **−8.2 (−10.8, −5.6)** | **−8.5 (−11.2, −5.8)** |
| Persons per bedroom | NA | NA |
| Anti-anxiety, antidepression or anti-insomnia medications: | | |
| No | Referent | Referent |
| Yes | **9.7 (6.7, 12.8)** | **12.2 (9.0, 15.4)** |
| Subjective health: | | |
| Poor/fair/good | Referent | Referent |
| Very good/excellent | **−11.2 (−13.4, −8.9)** | **−15.1 (−17.0, −13.1)** |
| Postal District: | | |
| Central | Referent | Referent |
| Eastern | **−4.0 (−6.7, −1.2)** | NA |
| Metropolitan Toronto | −2.0 (−4.6, 0.8) | |
| Northern | **−5.2 (−9.7, −0.6)** | |
| Southwestern | **−5.1 (−7.7, −2.6)** | |
| COVID-19 vaccine in last six months: | | |
| None | Referent | Referent |
| One or more | **1.5 (0.4, 2.5)** | **1.2 (0.3, 2.1)** |
| COVID-19 in last six months: | | |
| None | Referent | Referent |
| One or more | **4.0 (0.1, 8.0)** | NA |
| Occupation: | | |
| Teacher/instructor | Referent | Referent |
| Administrative/clerical[3] | **−8.0 (−12.7, −3.1)** | −3.7 (−8.6, 1.4) |
| Educational assistant/ECE | 0.1 (−3.5, 3.8) | **−3.7 (−7.2, −0.1)** |
| Other, operational support[4] | −3.9 (−10.5, 3.0) | **−8.6 (−14.8, −1.9)** |
| Principal/VP | −0.4 (−4.9, 4.4) | −3.9 (−8.4, 0.7) |
| Other, professional support[5] | 0.6 (−3.7, 5.1) | −1.8 (−6.0, 2.6) |

*(Continued)*

**Table 3.** (Continued)

| Variable | Anxiety score estimates[1] (95% CI) | Depression score estimates[2] (95% CI) |
|---|---|---|
| Highest level of student contact: | | |
| None | Referent | Referent |
| In room, often <2 metres | NA | NA |
| In room, rarely <2 metres | | |
| Physical contact | | |
| Highest level of coworker contact: | | |
| None | Referent | Referent |
| In room, often <2 metres | 1.9 (−0.5, 4.3) | NA |
| In room, rarely <2 metres | 0.8 (−1.5, 3.2) | |
| Physical contact | **4.4 (1.1, 7.8)** | |
| Number of student contacts per week | NA | NA |

Bold identifies group significantly different (p < 0.05) from referent group.

ECE: Early Childhood Educator; NA: not applicable (not included in model).

[1] Conditional $R^2$: 60.8%; Marginal $R^2$: 11.8%.

[2] Conditional $R^2$: 64.4%; Marginal $R^2$: 14.0%.

[3] office and clerical staff, superintendent, human resources, finance, planner.

[4] information technology, audio-visual or computer technicians, bus driver, custodian, building maintenance (carpenter, plumber), cafeteria staff, lunchroom assistant.

[5] psychologist, social worker, therapist, librarian, registered nurse.

weeks of returning to work [49]. Longer holidays (>14 days) have also been shown to improve well-being, but levels of anxiety and depression returned to baseline within a week of resuming work [50].

While there is consensus that students learn better in the classroom than online [51], school re-openings were associated with higher education worker distress scores. The current study found that during the first month of in-person work after school lockdowns, education workers' distress scores were 4.4% higher than other periods. Increased anxiety, stress, and depression due to the return to face-to-face teaching were also noted in a Spanish study conducted in 2020 [13]. As well, in England, teachers' self-reported anxiety peaked in May 2020 when June school re-opening announcements were made [52]. Of note, data collection for this study started after the first two return-to-workplace periods in Ontario which may have attenuated the effect size for this estimate.

During the pandemic, educators coped with work and personal stress in myriad ways. In the current study, it is unknown if the 15.6% of education workers who indicated, at baseline, that they used anti-anxiety, antidepression or anti-insomnia medications reflects a pre-existing issue or a peri-pandemic stress reduction strategy. In a Jordanian study, 9.9% of university educators indicated that they used medications (anti-anxiety medications, opioid analgesics, sedative hypnotics, antipsychotics, or psychic stimulants) to manage their COVID-19 associated distress [18]. Thompson et al. found that 12% of the general population in eastern Canada who responded to their survey indicated that they were drinking more alcohol since the start of the pandemic and that drinking more frequently was associated with feelings of stress, loneliness, or hopelessness [53]. A 2021 report found that among Canadian respondents who consumed alcohol or cannabis, 24% and 34%, respectively, indicated that their consumption had increased during the pandemic [54]. A scoping review reported that mindfulness was the most popular intervention to reduce stress in teachers, with most studies reporting a positive result; Cognitive Behavioural Therapy also reduced perceived stress levels [55]. As well, teachers reported less mental distress if they felt supported by principals, school-based administrators, their school board, the Ministry of Education, their union and colleagues, and if recognized within and beyond their schools [56].

Excellent or very good subjective health was associated with lower K10 scores. An American study that used the six-item Kessler Psychological Distress Scale [29] found that physical activity was positively and directly associated with both subjective health and mental health [57]. Levels of physical activity changed during the pandemic. Using cell phone data, Tison et al. found that daily step counts dropped by 5.5% within 10 days of the WHO pandemic declaration and by 27.3% within 30 days [58] with declines in countries that had not instituted lockdowns. Early identification of those most likely to decrease their level of physical activity during stressful times can be used in messaging regarding how activity levels can be maintained in future periods of distress.

Study participants were self-selected and withdrew before the end of the study due to retirement, parental leave, or other reasons. As a result, while the COVID-19 Cohort Study participant characteristics were reflective of the Ontario teaching population [31], study findings may not be generalizable to all Ontario education workers. Further, COVID-19 policies and employment conditions varied across Canada [59]. Thus, findings may not be generalizable to all education workers. Social desirability may have biased self-reported symptoms. As no pre-pandemic K10 measures are available and data were not collected in 2020, this study cannot provide information to compare with those periods. However, the study has a large sample size that collected data for 34 consecutive months using a validated distress scale.

## Conclusions

Early identification of people most likely to experience distress is needed so that stress remediation strategies can be quickly implemented. All workers should be offered evidence-informed distress lowering strategies in stressful situations. Younger, female education workers with lower rated subjective health, who are taking medications to reduce anxiety, depression, or sleeping problems are likely to be at the highest risk. Recognition of periods that intensify already stressful situations, such as pandemics and phases within them (e.g., returning to in-person work or higher mitigation based on higher rates of COVID-19 illness), is important to anticipate the need for assistance. Future pandemics, health emergencies, and other events that lead to education service disruptions need to be met with education systems and workers who can continue to provide an exceptional learning experience. In addition to further longitudinal research into best practices in early distress identification and remediation, a comprehensive, destigmatizing mental health framework targeted to the needs of all education workers that recognises periods of stress intensification as well as high-risk demographic characteristics that can be quickly implemented across school districts and across educational institutions is needed.

## Supporting information

**S1 Table. Percent change in K10 scores per four weeks, Ontario education workers participating in the K10 substudy with five or more submissions, February 18, 2021 to December 22, 2023, linear mixed effects model.** (PDF)

## Acknowledgments

The investigators thank their staff, who worked tirelessly throughout the studies and the participants, who gave freely of their time throughout this stressful pandemic.

## Author contributions

**Conceptualization:** Iris Gutmanis, Brenda L. Coleman, Robert Maunder, Allison McGeer.

**Data curation:** Brenda L. Coleman, Iris Gutmanis.

**Formal analysis:** Brenda L. Coleman, Kelly Ramsay.

**Funding acquisition:** Brenda L. Coleman, Allison McGeer.

**Methodology:** Brenda L. Coleman.

**Project administration:** Brenda L. Coleman.

**Resources:** Brenda L. Coleman, Allison McGeer.

**Supervision:** Brenda L. Coleman.

**Validation:** Brenda L. Coleman.

**Writing – original draft:** Brenda L. Coleman, Iris Gutmanis.

**Writing – review & editing:** Brenda L. Coleman, Iris Gutmanis, Kelly Ramsay, Robert Maunder, Susan J Bondy, Kailey Fischer, Veronica Zhu, Allison McGeer.

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
