## [Decision Letter · Decision Letter 0]

5 Jan 2025

PONE-D-24-47690Changes in emotional distress among Ontario education workers during the COVID-19 pandemicPLOS ONE

Dear Dr. Coleman,

Thank you for submitting your manuscript to PLOS ONE. After careful consideration, we feel that it has merit but does not fully meet PLOS ONE’s publication criteria as it currently stands. Therefore, we invite you to submit a revised version of the manuscript that addresses the points raised during the review process.

We look forward to receiving your revised manuscript.

Kind regards,

Tauseef Ahmad, PhD

Academic Editor

PLOS ONE

Journal Requirements:

Additional Editor Comments:

The authors have presented some very interesting points in the manuscript, but these sections require revision for clarity and accuracy.

Consider providing more detailed background information about the existing literature. For example, how does this study build upon prior research on the psychological impacts of COVID-19 on educators or other frontline workers?

Clarify the rationale for focusing specifically on Ontario education workers. Is there something unique about this population that warrants investigation?

Provide a clear statement of the research question or hypothesis.

The inclusion criteria mention education workers aged 18-77 who were employed in Ontario schools and working an average of ≥8 hours/week, but the exclusion criteria for those who selected "neither male nor female" as their gender are vague. Clarify why participants who selected "neither male nor female" were excluded due to "small sample size." Providing more context on this decision, such as whether the data from non-binary participants was insufficient to draw meaningful conclusions, would make this clearer. Additionally, consider expanding on the inclusion and exclusion criteria in more detail for transparency, particularly if gender diversity is a relevant aspect to the study.

The inclusion of COVID-19-specific factors (e.g., infection history, vaccination status) is mentioned, but their relationship to K10 scores is not fully explored or explained in the methodology.

Report any statistical significance (e.g., p-values or confidence intervals) for the findings related to K10 scores and associated factors. For example, was the decrease in K10 scores over time statistically significant?

The relationship between specific periods (e.g., "return-to-workplace periods") and distress could be elaborated further. How much variation in K10 scores was explained by these factors?

Reviewers' comments:

Reviewer's Responses to Questions

**Comments to the Author**

1. Is the manuscript technically sound, and do the data support the conclusions?

Reviewer #1: Yes

2. Has the statistical analysis been performed appropriately and rigorously? 

Reviewer #1: Yes

3. Have the authors made all data underlying the findings in their manuscript fully available?

Reviewer #1: Yes

4. Is the manuscript presented in an intelligible fashion and written in standard English?

Reviewer #1: Yes

5. Review Comments to the Author

Reviewer #1: I enjoyed reading your article and found the methods generally rigorous and the findings interesting. There were a few places where I wished for some clarification to understand what was done and how better :

1) Is there a particular reason why you chose these specific start and end-dates; perhaps especially the end-dates?

2) I don't understand the mitigation scoring and variables. Would it be possible to explain what it is based on more clearly?

3) With respect to the four-week intervals respecting the K-10 scores, I think I understand what you did based on the elaboration in the results, but here too I think the methodology could be described more clearly.

6. PLOS authors have the option to publish the peer review history of their article (what does this mean? ). If published, this will include your full peer review and any attached files.

**Do you want your identity to be public for this peer review?** For information about this choice, including consent withdrawal, please see our Privacy Policy .

Reviewer #1: No

---

## [Author Response · Author response to Decision Letter 1]

11 Feb 2025

Responses to all the reviewer and editor comments were uploaded as the file "Response to Reviewers"

Response to Reviewers

We thank the reviewers for their comprehensive reviews. Our responses are captured under each comment. Please note that when lines are referenced, we are referring to the manuscript that highlights changes to the original version, the file labelled “Revised Manuscript with Track Changes”.

Editor comments

We have reviewed this submission and ensured that we are compliant with the PLOS ONE style templates.

2. We note that you have indicated that there are restrictions to data sharing for this study. Before we proceed with your manuscript, please address the following prompts:

If there are ethical or legal restrictions on sharing a de-identified data set, please explain them in detail (e.g., data contain potentially identifying or sensitive patient information, data are owned by a third-party organization, etc.) and who has imposed them (e.g., a Research Ethics Committee or Institutional Review Board, etc.). Please also provide contact information for a data access committee, ethics committee, or other institutional body to which data requests may be sent.

The following was stated in the original submission:

The datasets generated and/or analysed during the current study are not publicly available due to information that could compromise the privacy of research participants but are available from the corresponding author on reasonable request.

This has been replaced with:

The datasets generated and/or analysed during the current study are not publicly available as the data file contains potentially identifying patient information. Requests for data access should be sent to:

Dr. Brenda L. Coleman, PhD

Clinical Scientist, Sinai Health, Toronto

Assistant Professor, Dalla Lana School of Public Health, University of Toronto

b.coleman@utoronto.ca

3. PLOS requires an ORCID iD for the corresponding author in Editorial Manager on papers submitted after December 6th, 2016. Please ensure that you have an ORCID iD and that it is validated in Editorial Manager.

The following information has been added: Dr. Brenda Coleman: 0000-0002-7144-4827

4. Please review your reference list to ensure that it is complete and correct.

The reference list has been reviewed to ensure compliance with PLOS ONE submission requirements.

Additional Editor Comments:

1. Consider providing more detailed background information about the existing literature. For example, how does this study build upon prior research on the psychological impacts of COVID-19 on educators or other frontline workers?

Additional information from longitudinal studies has been added (see lines 84-99).

2. Clarify the rationale for focusing specifically on Ontario education workers. Is there something unique about this population that warrants investigation?

Additional information regarding the importance of this group of essential workers has been added to the introduction (see lines 59-65). In addition, details regarding the generalizability of this information have also been added to the limitations section (see lines 379-381).

3. Provide a clear statement of the research question or hypothesis.

The study hypothesis is now stated (see lines 105-109).

4. The inclusion criteria mention education workers aged 18-77 who were employed in Ontario schools and working an average of ≥8 hours/week, but the exclusion criteria for those who selected "neither male nor female" as their gender are vague. Clarify why participants who selected "neither male nor female" were excluded due to "small sample size." Providing more context on this decision, such as whether the data from non-binary participants was insufficient to draw meaningful conclusions, would make this clearer. Additionally, consider expanding on the inclusion and exclusion criteria in more detail for transparency, particularly if gender diversity is a relevant aspect to the study.

More detail has now been provided regarding the decision to drop those who did not select male or female as their gender (see lines 123-126).

5. The inclusion of COVID-19-specific factors (e.g., infection history, vaccination status) is mentioned, but their relationship to K10 scores is not fully explored or explained in the methodology.

More details regarding how variables were created are now provided in the section “Survey instruments” (see lines 139-173).

6. Report any statistical significance (e.g., p-values or confidence intervals) for the findings related to K10 scores and associated factors. For example, was the decrease in K10 scores over time statistically significant?

The relationship between K10 scores and the four temporal variables was statistically significant (see lines 234-240). Other significant factors are described in lines 249-254. Further, as indicated in the notes for Table 2, all factors significantly associated with K10 distress scores are bolded.

7. The relationship between specific periods (e.g., “return-to-workplace periods”) and distress could be elaborated further. How much variation in K10 scores was explained by these factors?

Details of the relationship between the temporal variables and distress are provided in lines 240-246. The model conditional R2 and the marginal R2 are reported in the Table 2 notes and now also on lines 254-257.

8. While revising your submission, please upload your figure files to the Preflight Analysis and Conversion Engine (PACE) digital diagnostic tool.

The single figure that accompanies this paper has be uploaded to the PACE digital diagnostic tool.

Reviewer comments

1. Is the manuscript technically sound, and do the data support the conclusions?

Reviewer #1: Yes

2. Has the statistical analysis been performed appropriately and rigorously?

Reviewer #1: Yes

3. Have the authors made all data underlying the findings in their manuscript fully available?

Reviewer #1: Yes

4. Is the manuscript presented in an intelligible fashion and written in standard English?

Reviewer #1: Yes

5. Review Comments to the Author

Reviewer #1: I enjoyed reading your article and found the methods generally rigorous and the findings interesting. There were a few places where I wished for some clarification to understand what was done and how better:

1) Is there a particular reason why you chose these specific start and end-dates; perhaps especially the end-dates?

The study was started once funding was obtained. Initially, funding was requested for 14 months. Then, we applied for and received additional funding to continue the study until the end of 2023. New funding was not requested for the 2024 calendar year (see lines 115-122).

2) I don't understand the mitigation scoring and variables. Would it be possible to explain what it is based on more clearly?

Additional information has now been provided (see lines 168-173)

3) With respect to the four-week intervals respecting the K-10 scores, I think I understand what you did based on the elaboration in the results, but here too I think the methodology could be described more clearly.

We hope the new wording is clearer (see lines 157-158).

6. PLOS authors have the option to publish the peer review history of their article. Do you want your identity to be public for this peer review? For information about this choice, including consent withdrawal, please see our Privacy Policy.

Reviewer #1: No

We thank the reviewers for their time and expertise.

Regards,

Dr. Brenda L. Coleman, PhD

Clinical Scientist, Sinai Health, Toronto

Assistant Professor, Dalla Lana School of Public Health, University of Toronto

Sinai Health System

600 University Ave

Toronto, ON, Canada M5G 1X5

(C) 647-267-2413

b.coleman@utoronto.ca

---

## [Decision Letter · Decision Letter 1]

10 Apr 2025

PONE-D-24-47690R1Changes in emotional distress among Ontario education workers during the COVID-19 pandemicPLOS ONE

Dear Dr. Coleman,

Thank you for submitting your manuscript to PLOS ONE. After careful consideration, we feel that it has merit but does not fully meet PLOS ONE’s publication criteria as it currently stands. Therefore, we invite you to submit a revised version of the manuscript that addresses the points raised during the review process.

We look forward to receiving your revised manuscript.

Kind regards,

Tauseef Ahmad, PhD

Academic Editor

PLOS ONE

Journal Requirements:

**Additional Editor Comments:**

The manuscript requires minor revisions before proceeding further. The reviewer has provided the following comment: "The work-related stress during the COVID-19 pandemic was common among all types of workers across the globe. This paper focuses only on education sector workers. While this cohort study is interesting and useful, it could be more meaningful if the authors include literature on work-related stress among workers in other sectors, particularly the healthcare sector." Addressing this point will enhance the comprehensiveness of the study.

Reviewers' comments:

Reviewer's Responses to Questions

**Comments to the Author**

1. If the authors have adequately addressed your comments raised in a previous round of review and you feel that this manuscript is now acceptable for publication, you may indicate that here to bypass the “Comments to the Author” section, enter your conflict of interest statement in the “Confidential to Editor” section, and submit your "Accept" recommendation.

Reviewer #1: All comments have been addressed

Reviewer #2: All comments have been addressed

2. Is the manuscript technically sound, and do the data support the conclusions?

Reviewer #1: Yes

Reviewer #2: Partly

3. Has the statistical analysis been performed appropriately and rigorously? 

Reviewer #1: Yes

Reviewer #2: No

4. Have the authors made all data underlying the findings in their manuscript fully available?

Reviewer #1: No

Reviewer #2: Yes

5. Is the manuscript presented in an intelligible fashion and written in standard English?

Reviewer #1: Yes

Reviewer #2: Yes

6. Review Comments to the Author

Reviewer #1: (No Response)

Reviewer #2: The work-related stress during COVID-19 pandemic was common among all types of workers across the globe. This paper talks only about the education sector workers. Though this cohort study is interesting and useful, it could be more meaningful if the authors add some literature reviews about the other sector workers particularly health sector workers.

7. PLOS authors have the option to publish the peer review history of their article (what does this mean? ). If published, this will include your full peer review and any attached files.

**Do you want your identity to be public for this peer review?** For information about this choice, including consent withdrawal, please see our Privacy Policy .

Reviewer #1: No

Reviewer #2: **Yes: ** Dr. Jayanta Kuar Basu

---

## [Author Response · Author response to Decision Letter 2]

14 May 2025

April 24, 2025

RE: PONE-D-24-47690R1

Changes in emotional distress among Ontario education workers during the COVID-19 pandemic: 2021-2023, PLOS ONE submission

Response to Reviewers

We thank the reviewers for their comprehensive reviews. Our responses are captured under each comment. Please note that when lines are referenced, we are referring to the manuscript that highlights changes to the original version, the file labelled “Revised Manuscript with Track Changes”.

Editor Comments

The manuscript requires minor revisions before proceeding further. The reviewer has provided the following comment: "The work-related stress during the COVID-19 pandemic was common among all types of workers across the globe. This paper focuses only on education sector workers. While this cohort study is interesting and useful, it could be more meaningful if the authors include literature on work-related stress among workers in other sectors, particularly the healthcare sector." Addressing this point will enhance the comprehensiveness of the study.

Literature regarding work-related stress among workers in other sectors has been added to the discussion. Please see lines 321-341 in the track changes copy of the resubmitted manuscript.

Reviewers' comments:

Reviewer's Responses to Questions: Comments to the Author

1. If the authors have adequately addressed your comments raised in a previous round of review and you feel that this manuscript is now acceptable for publication, you may indicate that here to bypass the “Comments to the Author” section, enter your conflict of interest statement in the “Confidential to Editor” section, and submit your "Accept" recommendation.

Reviewer #1: All comments have been addressed

Reviewer #2: All comments have been addressed

2. Is the manuscript technically sound, and do the data support the conclusions?

Reviewer #1: Yes

Reviewer #2: Partly

Additional information regarding the impact of the COVID-19 pandemic on workers in other sectors has been added. Please see lines 321-341 in the track changes copy.

3. Has the statistical analysis been performed appropriately and rigorously?

Reviewer #1: Yes

Reviewer #2: No

The Reviewer has not provided any details regarding what aspect of the statistical analysis was not performed appropriately. As no reviewer comments were submitted regarding this issue either in the first or second review, no changes have been made to the manuscript.

4. Have the authors made all data underlying the findings in their manuscript fully available?

The PLOS Data policy requires authors to make all data underlying the findings described in their manuscript fully available without restriction, with rare exception (please refer to the Data Availability Statement in the manuscript PDF file). The data should be provided as part of the manuscript or its supporting information or deposited to a public repository. For example, in addition to summary statistics, the data points behind means, medians and variance measures should be available. If there are restrictions on publicly sharing data—e.g. participant privacy or use of data from a third party—those must be specified.

Reviewer #1: No

Reviewer #2: Yes

The authors believe that his issue has already been addressed.

5. Is the manuscript presented in an intelligible fashion and written in standard English?

Reviewer #1: Yes

Reviewer #2: Yes

6. Review Comments to the Author

Reviewer #1: (No Response)

Reviewer #2: The work-related stress during COVID-19 pandemic was common among all types of workers across the globe. This paper talks only about the education sector workers. Though this cohort study is interesting and useful, it could be more meaningful if the authors add some literature reviews about the other sector workers particularly health sector workers.

Additional information regarding the impact of the COVID-19 pandemic on workers in other sectors has been added. Please see lines 321-341 in the track changes copy.

7. PLOS authors have the option to publish the peer review history of their article (what does this mean?). If published, this will include your full peer review and any attached files. Do you want your identity to be public for this peer review? For information about this choice, including consent withdrawal, please see our Privacy Policy.

Reviewer #1: No

Reviewer #2: Yes: Dr. Jayanta Kuar Basu

We thank the reviewers for their time and expertise.

If there are any further questions regarding this manuscript, please don’t hesitate to be in touch.

Regards,

Dr. Brenda L. Coleman, PhD

Clinical Scientist, Sinai Health, Toronto

Assistant Professor, Dalla Lana School of Public Health, University of Toronto

Sinai Health System

600 University Ave

Toronto, ON, Canada M5G 1X5

(C) 647-267-2413

b.coleman@utoronto.ca

---

## [Decision Letter · Decision Letter 2]

22 Jul 2025

PONE-D-24-47690R2Changes in emotional distress among Ontario education workers during the COVID-19 pandemic: 2021-2023PLOS ONE

Dear Dr. Coleman,

Thank you for submitting your manuscript to PLOS ONE. After careful consideration, we feel that it has merit but does not fully meet PLOS ONE’s publication criteria as it currently stands. Therefore, we invite you to submit a revised version of the manuscript that addresses the points raised during the review process.

We look forward to receiving your revised manuscript.

Kind regards,

Ali Cetin

Academic Editor

PLOS ONE

Journal Requirements:

Additional Editor Comments :

Dr. Brenda L. Coleman, Ph.D.

Corresponding Author

RE: Minor Revisions for Manuscript PONE-D-24-47690R2, "Changes in emotional distress among Ontario education workers during the COVID-19 pandemic: 2021-2023"

Dear Dr. Coleman,

Thank you for your resubmission of the manuscript referenced above to PLOS ONE. We commend you and your co-authors for the thorough and thoughtful revisions, which have substantially strengthened the manuscript and successfully addressed the primary concerns raised by the reviewers. The addition of comparative literature for other essential worker sectors has significantly enhanced the context and impact of your findings.

The manuscript is now very close to being suitable for publication. To further elevate its contribution, we recommend a final set of minor revisions focused on deepening the interpretation and sharpening the conclusions. Please find our specific suggestions below.

List of Recommended Revisions

1. Deepen the Interpretive Analysis in the Discussion Section

Location: In the Discussion section, as a new paragraph following the comparison with healthcare and other essential workers (i.e., after line 341 in the track-changes manuscript).

Rationale: To move beyond reporting the comparison and interpret why educators might have experienced higher levels of distress, thereby strengthening the manuscript’s analytical contribution.

Current Text (end of the paragraph):

"...lower than what was found in the current study (2021: 54.7%; 2023: 32.5%) but, in both cases, much higher than what is expected in the general population [44]."

Proposed Text (with new paragraph added):

"...lower than what was found in the current study (2021: 54.7%; 2023: 32.5%) but, in both cases, much higher than what is expected in the general population [44].

The finding that education workers, at times, reported higher levels of distress than even frontline healthcare providers warrants further consideration. This discrepancy may be attributable to several factors unique to the education sector. Unlike healthcare professionals, who are trained for crisis response, educators were thrust into a public health crisis with little to no specific preparation, while simultaneously managing the pedagogical, emotional, and safety needs of children. Furthermore, the intense public and political scrutiny surrounding school closures and reopenings placed educators at the center of a contentious societal debate. This confluence of responsibilities, acting as de facto frontline workers while maintaining educational continuity and navigating significant public pressure, likely created a unique and potent combination of stressors not experienced in the same way by other worker groups."

2. Strengthen the Policy Implications in the Conclusion Sub-section

Location: At the end of the "Conclusions" section, following the final sentence (after line 409).

Rationale: To provide a clear, actionable takeaway that highlights the practical and policy relevance of the research findings.

Current Text (final sentence):

"Further research into best practices in early distress identification and remediation is warranted to ensure future pandemics are met with education systems and workers who can continue to provide an exceptional learning experience ."

Proposed Text (with new sentences added):

"Further research into best practices in early distress identification and remediation is warranted to ensure future pandemics are met with education systems and workers who can continue to provide an exceptional learning experience .

Therefore, these findings underscore the need for education authorities to move beyond reactive measures and develop proactive, institutional-level mental health frameworks. Such frameworks should include targeted support for high-risk demographics and provide evidence-based strategies to manage workload and mitigate distress during future public health crises or other systemic disruptions."

3. Enhance the Link Between Data (Figure 1) and Interpretation in the Discussion Section

Location: In the Discussion section, within the paragraph discussing the effect of summer vacations (starts around line 345).

Rationale: To explicitly connect the striking visual data from Figure 1 to the interpretation, reinforcing the argument about the impact of acute environmental factors.

Current Text:

"In the current study, distress levels were lower during Ontario education worker vacation periods (summers). This corroborates previous research that reported a strong, but limited, positive effect of vacations on mood [46] ."

Proposed Text (with new sentence integrated):

"In the current study, distress levels were lower during Ontario education worker vacation periods (summers).

The pronounced fluctuations observed in Figure 1, particularly the sharp decrease in distress during summer and the subsequent spike upon returning to the workplace, highlight the significant and immediate impact of work-related environmental factors. This corroborates previous research that reported a strong, but limited, positive effect of vacations on mood [46] ."

Please consider the specific phrasing provided in these recommendations as illustrative examples designed to guide your revisions. We encourage you to adapt this guidance in a manner that best aligns with your own authorial voice and the overall narrative of the manuscript.

We believe that incorporating these minor revisions will make your manuscript an even more impactful and compelling contribution to the field. We look forward to receiving the revised version.

Sincerely,

The Editor

PLOS ONE

Reviewers' comments:

Reviewer's Responses to Questions

**Comments to the Author**

1. If the authors have adequately addressed your comments raised in a previous round of review and you feel that this manuscript is now acceptable for publication, you may indicate that here to bypass the “Comments to the Author” section, enter your conflict of interest statement in the “Confidential to Editor” section, and submit your "Accept" recommendation.

Reviewer #1: All comments have been addressed

Reviewer #2: All comments have been addressed

2. Is the manuscript technically sound, and do the data support the conclusions?

Reviewer #1: Yes

Reviewer #2: Yes

3. Has the statistical analysis been performed appropriately and rigorously? 

Reviewer #1: Yes

Reviewer #2: N/A

4. Have the authors made all data underlying the findings in their manuscript fully available?

Reviewer #1: Yes

Reviewer #2: Yes

5. Is the manuscript presented in an intelligible fashion and written in standard English?

Reviewer #1: Yes

Reviewer #2: Yes

6. Review Comments to the Author

Reviewer #1: (No Response)

Reviewer #2: All concerns were addressed, and necessary justification was provided. The authors are suggested to try similar studies with other sector workers and do a comparative analysis.

7. PLOS authors have the option to publish the peer review history of their article (what does this mean? ). If published, this will include your full peer review and any attached files.

**Do you want your identity to be public for this peer review?** For information about this choice, including consent withdrawal, please see our Privacy Policy .

Reviewer #1: No

Reviewer #2: **Yes: ** Dr. JAYANTA KUMAR BASU

---

## [Author Response · Author response to Decision Letter 3]

30 Jul 2025

Our response to the reviewer is attached as a separate file.

---

## [Editor Report · Decision Letter 3]

1 Aug 2025

Changes in emotional distress among Ontario education workers during the COVID-19 pandemic: 2021-2023

PONE-D-24-47690R3

Dear Dr. Coleman,

We’re pleased to inform you that your manuscript has been judged scientifically suitable for publication and will be formally accepted for publication once it meets all outstanding technical requirements.

Kind regards,

Ali Cetin

Academic Editor

PLOS ONE

Additional Editor Comments (optional):

I think you answered all the comments.
---

## [Editor Report · Acceptance letter]

PONE-D-24-47690R3

PLOS ONE

Dear Dr. Coleman,

I'm pleased to inform you that your manuscript has been deemed suitable for publication in PLOS ONE. Congratulations! Your manuscript is now being handed over to our production team.

Kind regards,

on behalf of

Professor Ali Cetin

Academic Editor

PLOS ONE